# Management and Organizational Research: Structural Topic Modeling for a Better Understanding of Theory Application

**Rohit Bhuvaneshwar Mishra *** and Hongbing Jiang *

School of Management Engineering, Zhengzhou University, Zhengzhou 450001, China
* Correspondence: rohit.bnmishra123@gmail.com (R.B.M.); jhbymx@zzu.edu.cn (H.J.)

**Abstract:** In management and organization research, theory development is often linked with developing a new theory. However, regardless of the number of existing theories, most theories remain empirically untested, and the progress in understanding the application of theories has been scarce. This article discusses how theories are applied in existing management and organization research studies. This study applies the Structural Topic Model to 4636 research papers from the S2ORC dataset. The results reveal twelve research themes, establish correlations, and document the evolution of themes over time. The findings of this study reveal that the theoretical application is not consistent across research themes, theories are primarily used for descriptive and communicative properties, and most research themes in management and organization research are more concerned with discovering phenomena rather than with understanding and forecasting them.

**Keywords:** management and organization theory; management science; theory research; structural topic modeling; data and text analysis

## 1. Introduction

A theory is a collection of constructs linked together by logically consistent propositions [1]. A theory is the most essential and fundamental building component in an academic study [2,3]. The theory is nevertheless related to "what", "how", "when", and "why" elements, interpreting the happenings of the world and predicting them [4–6]. Any cohesive explanation or description of seen, experienced, or recorded occurrences might be considered a theory [7]. Although some methods of descriptive analysis are often equated to theory, all scholars agree that categorizing data—qualitative or quantitative—is not theory [1]. Simply put, although empirical research would establish the occurrence of related events, theory explains why they happen [8].

Theories are extremely valuable instruments that allow us to achieve several significant results in an academic subject of study [9]. First, theories organize and promote knowledge. Second, theoretical understanding elucidates the links and interrelationships between individuals, teams, and organizations and help us to produce more accurate predictions and expectancies about individuals, teams, and organizations [10]. Third, theory guides research and provides a context for researchers [11,12]. Finally, theory enables us to understand isolated components of phenomena and offers a framework for focusing attention on certain facets [13]. Thus, theory allows us to understand the application regions, underlying assumptions, and the types of constructs to interact with [8,14,15].

In management and organization research, theory development is a significant task. Numerous studies have explored the structure of theory, philosophical concerns, theory kinds, constructs, scope, concept, and derived terms [1,6,7,15–21]. However, theory building is often associated with the construction of new theories. This approach has resulted in the explosion of theories, the majority of which remain empirically untested [22]. We would make more incredible progress in management and organizational theory development if we paid more attention to how existing theories are utilized in academic research [22,23].

This article sheds light on this fundamental concern about using theory in the management and organization discipline to further theoretical advancements in management and organizational research. We investigate the application themes of theories, find out how these themes are related to each other, and the possible research gaps and future areas for research using theories. This study aims to identify areas that have received sufficient and insufficient research attention in previous studies. We evaluate the following research questions (RQ) in light of this background and motivation:

- RQ 1. What are the research topics using theories in Management and Organization Research?
- RQ 2. How are these research topics correlated?
- RQ 3. How have these research topics progressed with time?

Our objective is to analyze the condition of theoretical diversity, not to compile an exhaustive list of ideas employed using management and organization theories. Instead, we investigate the use of theory in articles to address these problems. Numerous stakeholders benefit from greater knowledge of theory application. For example, students can use knowledge of theory application to determine the validity and distinctiveness of their research topics. Additionally, academic publishers' review teams can use the theory application knowledge to understand which theories are typically used in a certain research stream and how to evaluate their application in a particular academic work [24,25]. Thus, understanding how theories are applied in research will result in a greater academic understanding of diversity in management and organizational research. Additionally, it will enrich an in-depth analysis of the discipline's intellectual structure from a theory-usage perspective, such as in particular streams of research within the discipline.

We use the computational technique of structural topic modeling [26,27] to rigorously study the distribution of theory usage while also visualizing the relationships between different research themes. These themes are not pre-determined for the model but develop inductively when the algorithm identifies patterns within the texts. To maintain the fairness of our analysis, we employ a clear meaning of theory consistent with [1,6,15,16,18–20], which defines theory as "anything which analyzes, predicts or explains phenomena". Consistent with this definition of theory, we consider an article to have used a theory to "analyze, predict, or explain" a phenomenon if it explicitly uses a theory name in the abstract text. Then, we examine word co-occurrence patterns in the abstract text. Finally, topic modeling analysis entails a relationship-based interpretation of word meaning. As a result, the model is capable of doing an exploratory study on datasets that are too large for human encoding. Machine learning is used to uncover patterns and correlations that may have gone unnoticed during human coding or conventional text analysis. The study's findings are then addressed, focusing on the implications for defining how theories should be used in the management and organization disciplines. We conclude by summarizing our results, highlighting their limitations, and outlining future research directions. The research design is in the Figure 1 below:

- Data Collection
    - S2ORC Dataset
    - Search Terms: 40 management and organization theories

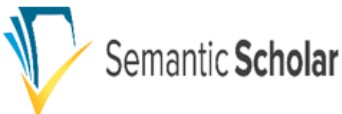

- Data Preparation
    - Preprocessing
    - Removing instances of word "theory"

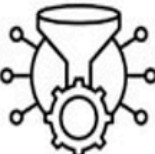

- Model Training
    - Training abstract text with STM model
    - Meta data covariate: Publication date of research article

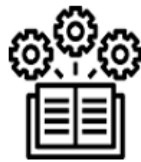

- Analysis
    - Topic identification
    - Correlation between topics
    - Impact of covariate on topic prominence

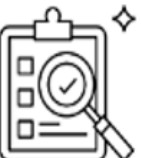

**Figure 1.** Research Design.

## 2. Literature Review

Theories are built in several ways, including qualitative descriptions of events [28], via the interconnection of assumptions [29], and even by the use of imaginations [30]. Whatever method is used to devise a theory, theorists have a similar goal: to explain the world [4–6,20]. Numerous perspectives exist to assist theorists in generating theories that adhere to theoretical goals such as validity, usefulness, falsifiability, and parsimony. Above all, theory seeks to clarify notions for communication, provide the intellectual foundation for comprehension and discussion about the world described by the theory, and describe the nature of the interactions that result in enhanced knowledge.

There have been many scientists who claim that research should be theory-based [1,30]. However, management and organization research depends on a diverse range of theories [9,31,32]. Therefore, to comprehend management and organization study, one needs to learn about the most important theories in the domain and their application areas. This perspective is particularly relevant considering the ever-increasing complexity and pace of happenings in management and organizational study.

There have been various qualitative studies on theories in management and organization research. For example, Bacharach [1] developed a framework for assessing theories by examining the falsifiability and usefulness of variables, constructs, and relations. Lee [23] conducted a study on qualitative methods in accounting and management research. According to Edwards [22], management and organizational research should place a greater emphasis on theory refinement, which includes identifying the limits of theories, orchestrating tests between competing theories, and increasing the exactness of theories to produce strong predictions. Cheng [33] reviewed international management studies and argued

that the present method, which is theory-driven and phenomenon-based, perceives country settings as limits and is based on data from a particular area. Cheng argues this is a significant limitation of the current approach and it is hindering theory development. Zahra and Newey [32] presented three types of theory development and emphasized their strengths and drawbacks via the use of an "impact wheel" that visualizes their effect across five domains: "theory, field, discipline, researchers, and external stakeholders". Bluhm et al. [34] conducted an examination of qualitative research published on both sides of the Atlantic, highlighting the lessons that each can learn from the other. They listed notable papers and methodologies and made recommendations for the next decade of qualitative management research. Johnson [35] discussed how qualitative management and organizational research are evaluated. Duberley [36] suggested that maintaining the credibility of qualitative management research is necessary. Duberley re-emphasized the link between "epistemology and methodology" and recognized that seemingly similar methodology could be underpinned by very different knowledge-constructing assumptions. Cornelissen [37] addressed the distinctions between the many forms of theoretical explanations linked with quantitative and qualitative approaches, discussed the tendency toward a quantitative "restyling" of qualitative research, and focused on the negative consequences for body of knowledge and the condition of management and organization theory.

There have been a few quantitative studies also, but most have been descriptive, using simple analyses such as author distribution, citation analysis, or journal location [21,38–41]. For example, Miner [24] evaluated 32 proven management and organization theories based on their perceived significance, validity, and utility, and little evidence of any association between these three factors was found. Miner [25] conducted a more recent study in which he analyzed the rated importance, degree of recognition, validity, and utility of 73 established management and organization theories, distinguishing between the perspectives of judges with expertise in organizational behavior and strategic management. The findings of this study demonstrate that this science is maturing, and that many more positive correlations between the variables examined exist than was previously discovered. Colquitt and Zapata-Phelan [42] established a taxonomy that categorizes empirical publications according to their theoretical contribution on two dimensions: "theory building and theory testing". Based on a study of 74 issues of the "Academy of Management Journal", the results indicate that theory development and testing have increased over time. Additionally, the study shows that the extent to which papers construct and test theories is a strong predictor of citation rates. Jia [43] created a "context-emic model" to assess theoretical contributions to Chinese management research. Finally, some studies propose that theory development should be both theoretically and practically meaningful [3,44,45].

Although the studies stated above aid in various ways in comprehending theory, there is a dearth of research on the application of theories in management and organization. Additionally, academic research on relevant management and organization research themes has largely focused on developing hypotheses and applying theories to a variety of phenomena. By contrast, this article describes a method for automatically identifying research ideas based simply on the analysis of the research corpus. Despite previous studies that examined theories in a variety of ways, there is a gap in our understanding of how theories are used in academic research, what the various research themes of theory application across various streams of management and organization research are, how these themes relate to one another, and how these themes have evolved over time. As a result, we place emphasis on understanding how theory is applied, guided by the aforementioned research questions.

We use structural topic modeling (STM) in this study, a novel kind of topic modeling [26,27]. STM uses topics to reflect the overall structure of the corpus under examination [46–49]. In an article, a wide range of topics will be covered in varying degrees. Similarly, multiple articles will contain the same topic in different proportions [50–53]. In contrast to other types of topic modeling, STM explicitly assesses the effects of metadata on topic prevalence [54–56]. The findings illustrate patterns with which topics occur over time

and associations between variables and topic prevalence or word usage in a corpus. Put simply, STM is a better fit for our research than the more popular Latent Dirichlet allocation technique (LDA) [57]. STM is unusual because it incorporates variables at the document level whose distribution function may be influenced by covariate data [48,50,55].

In this research, we apply STM to a large corpus of research articles [58]. STM is an excellent tool for analyzing the relevant data set because it comprises the text of research articles as well as information that is conceptually and empirically connected to the narratives' contents. [58,59]. A detailed description of the dataset will be discussed later in this article. The methodologies outlined in this article can be utilized by analysts to discover relevant issues in management and organization research. Furthermore, the article demonstrates how the techniques used in this study, which have not been employed before in management and organization research, have the potential to determine future research objectives.

## 3. Methods

The study's strength stems from our use of novel computational tools. Automated text analysis provides several significant benefits over the qualitative approaches discussed earlier. When it comes to finding themes or performing any other categorization, topic modeling is the most useful method. It is structured, substantially convenient, quick, and includes ways for presenting methods and assertions in a clear and concise manner. Automated text analysis helps resolve issues related to excess, external comparisons, and opacity associated with a qualitative methodology. Quantitative text analysis enables qualitative researchers to analyze massive amounts of data without being overwhelmed by the volume of data available. The systematic connection of documents with metadata enables qualitative researchers to handle the preconceptions and chronological issues that often accompany external comparisons. Furthermore, quantitative text analysis will assist qualitative research in becoming more "transparent" by promoting data exchange, method openness, and sharing claims about its transparency.

The methods section is divided into three subsections. To begin, we will cover corpora, data collection, and processing in Section 3.1. Then, in Section 3.2, we will look at how to prepare data for structural topic modeling. Finally, in Section 3.3, we will discuss the validity of the output of a topic model.

### 3.1. Data Collection

The data were collected through a textual search that searched the abstracts of S2ORC research articles [58]. S2ORC is a vast corpus of English-language academic articles across a range of academic disciplines. The corpus contains extensive metadata, abstracts of articles, resolved bibliographic references, and entire formatted articles published in open access journals. The text is annotated with references to citations, images, and tables, all of which are linked to their corresponding paper objects. We have explicitly narrowed our search to articles from the S2ORC corpus' psychology papers. Automatically searching for theory names in the abstract text was accomplished by using 40 management and organization theory names as textual queries [9]. The theory names are in Table 1. Further, we filtered for articles with at least ten citations. Selecting articles with at least ten citations was performed to gather high-quality papers.

### 3.2. Data Preprocessing and Structural Topic Modeling

Preprocessing started with converting terms to lower text. Stop words such as "the", "a", and "an" were removed. Additionally, we eliminated all numbers, abbreviations, and punctuation. Furthermore, we stemmed the words to a standard "stem" for words (for example, "innovative" and "innovator") to have a single "stem" (for example, "innovate"). Due to the focus of our research on the application of theory, it is expected that the term "theory" would be used so many times in our corpus. Therefore, the term "theory" is predominated, but it does not add any information to our results. On the other hand, this

may block out other words, leading us to overlook important points in the text. Therefore, to make the data better for topic modeling, we deleted all instances of the term "theory" from the corpus. Then, we applied structural topic modeling to find the important topics, the correlation between topics, and the evolution of topics with time in the management and organization study corpus.

**Table 1.** Theory names used as textual query.

| 40 Management and Organization Theory Names as Textual Query | | | |
| --- | --- | --- | --- |
| Absorptive Capacity Theory | Actor-Network Theory | Agency Theory | Agenda Setting Theory |
| Attachment Theory | Attribution Theory | Balance Theory | Control Theory |
| Diffusion of Innovations Theory | Dynamic Capabilities Theory | Efficient Market Theory | Ethical Theory |
| Field Theory | Game Theory | Goal Setting Theory | Image Theory |
| Institutional Theory | Knowledge-Based Theory | Media Richness Theory | Mental Models Theory |
| Organizational Ecology Theory | Organizational Justice Theory | Planned Behavior Theory | Prospect Theory |
| Psychological Contract Theory | Resource-Based Theory | Role Theory | Self-Determination Theory |
| Sensemaking Theory | Social Capital Theory | Social Cognitive Theory | Social Comparison Theory |
| Social Exchange Theory | Social Facilitation Theory | Social Identity Theory | Social Network Theory |
| Stakeholder Theory | Structural Contingency Theory | Structuration Theory | Transaction Cost Theory |

Data collection from the S2ORC dataset was performed on Jupyter Notebooks using the python programming language. Further data processing, structural topic modeling, and visualization were performed on R studio using the R programming language.

In topic modeling, texts are organized into themes, which are then referred to as "topics" [48,53,55]. A topic is a word distribution that indicates interrelated themes in a text [50]. Topic models are well-suited for processing vast amounts of text data. Topics provide meaning to a group of words. We used a technique known as structural topics modeling for the topic modeling in this article [26,27]. In addition to topic modeling, STM allows us to find correlations among topics [50,54]. Covariate variables, such as document metadata, have an effect on the topical prominence of documents [48,60]. Furthermore, covariate data is also used to build word usage patterns within a topic. We utilized the date of publication of research articles from the S2ORC dataset as a metadata covariate. This is critical for comprehending how the narrative and topic's relative weights vary over time. Metadata may have an effect on a topic's prominence and substance. When metadata contains variables for topic prevalence, the metadata can have an effect on topic frequency. Similarly, alterations in topical content enable metadata to alter the frequency with which specific terms are used within a topic or the manner in which a topic is presented.

Due to the fact that STM is an unsupervised method, the number of topics included in the analysis is crucial [48,55]. As a result, we assess a variety of models covering a range of five to fifty topics. These models were then evaluated qualitatively for their capacity to produce coherent themes [55]. Additionally, the number of topics selected was determined by our understanding of the data set. Our study, which was required to analyze the number of themes, was also influenced by comparable previous research that employed topic modeling to extract meaning from massive text samples [55]. However, our objective was to use topic modeling to address the research questions, not advance the STM methodology.

Validating the topic is crucial for evaluating whether the topic's true meaning and associated phrases correspond to the text's subjective meaning. This was accomplished with the use of methodological guidance from previous studies [48,50]. Prior understanding about why texts were created and what they aspired to achieve affected the topic interpretations considered. The majority of the content was written and consumed by researchers in the management and organization field, and this perception was utilized to determine the prevalence or lack of topics and phrases. The vast majority of topic headings were straightforward and required little interpretation. Thus, the subject was clearly defined, given the narrative's genre and the narrative's emphasis on research surrounding famous themes [48,50]. When there were differences in the themes, the writers worked together to settle them. Thus, we followed the best techniques of prior research. However, we did

not make any new advancements. Additionally, as with traditional content analysis, to validate topic models, we require qualitative evaluation, in which researchers evaluate the comprehensibility and comparative efficacy of models using their subject matter expertise and dataset relevance.

### 3.3. Validation of STM Model

It is critical to understand that STM is a kind of unsupervised learning technique. We have no exact way of verifying a topic model's output. A method of verifying the topic model is to exclude certain words or documents from the model creation process and then assess the likelihood of missing data using the different models. Although calculating direct likelihood is challenging, their estimates are plausible [61,62].

Another often-used technique is to assess the model using perplexity [57]. However, the conclusions from this technique will also vary depending on the quantity of vocabulary in a corpus and the degree to which words are "predictable" within the corpus, making it hard to make general judgments about what makes a high or low score. Recent research indicates that the eventual value of topic models is determined by their logic and importance to analysts [63]. The most reliable way to validate these results would be to perform trials with experts who evaluate the relevancy and consistency of various themes, such as those described in [63]. Thus, we utilized the topic model to present our results without explicitly attempting to validate their models, but rather enabling readers to independently examine and verify the findings. We made a point of comparing our findings to earlier research in order to verify our models.

## 4. Result and Analysis

This section analyzes data and develops answers in response to the research questions we developed. In Section 4.1, we describe the data result. Section 4.2 is about selecting the number of topics for the STM model. Section 4.3 answers our first research question by identifying research topics using theories in Management and Organization Research. Section 4.4 answers the second research question by finding the correlation between identified research topics. Finally, Section 4.5 answers our third research question by identifying the proportions for all the topics as a function of time.

### 4.1. Data Result

The textual query to search the management and organization theory names in the corpus returned 20,140 research articles. Further consideration of only the research articles with a minimum of ten citations obtained 4636 articles. We tabulated the research papers according to the publication year and later divided them into four groups for the analysis in Table 2. We use the abstract text of these 4636 articles as our dataset for the STM modeling.

**Table 2.** Articles with at least 10 citations.

| Year Group | ≤1990 | 1991–2000 | 2001–2010 | 2011–2020 | All |
|---|---|---|---|---|---|
| Number of Articles | 57 | 217 | 1345 | 3017 | 4636 |

### 4.2. Selecting the Number of Topics

When deploying STM, the first key parameter is establishing the topic count. Although there is no universal answer to this problem, one alternative is to consider the trade-off between semantic coherence and exclusivity [55]. Semantic coherence is characterized by the frequency with which a word and word combinations occur. Semantic coherence may aid in the avoidance of difficult-to-define situations for several reasons. For example, when words may be chained together, it is critical to verify their linked meaning. For example, the phrase "information" may be related to the term "systems", which is associated with the term "machine", but their contexts are different. In such a situation, information and machine may not be assigned to the same topic [64]. In principle, the semantic coherence of a model reduces as the number of subjects increases. Another critical parameter is exclusivity [65].

The exclusive words are more likely to occur explicitly on one topic than on others. For instance, the data may include a subject relating to "management and organization". If the term "management and organization" is often used in connection with a "topic A" but not in other contexts with other "topic B", this demonstrates the topic's exclusivity. Moreover, in principle, with a rise in the number of themes in a model, the model's overall exclusivity often increases as well [55].

Figure 2 illustrates the value of semantic coherence on the horizontal axis and exclusivity on the vertical axis when STM chooses topics from five and fifty in the specified dataset. The graph points to the outcome for different values of topics. The number of themes detected is shown on the label. For example, the twenty-fourth data point demonstrates a model's semantic coherence and exclusivity after identifying twenty-four distinct themes. Additionally, the observed exclusivity values ranging between nine and 10 give little information regarding the frequency with which terms appear in the data. Thus, rather than these values, the focus should be made on data point comparisons. The projected trends in semantic coherence and exclusivity, as well as the trade-off between them, are represented in Figure 2. There is no obvious cut-off point for the number of topics that should be selected. Numerous data points stand out, including the data points of eleven, twelve, fourteen, twenty-four, and twenty-six themes. Perhaps the most glaring instance is the case of the discovery of twelve themes. Hence, we conduct all our analyses with twelve themes in mind.

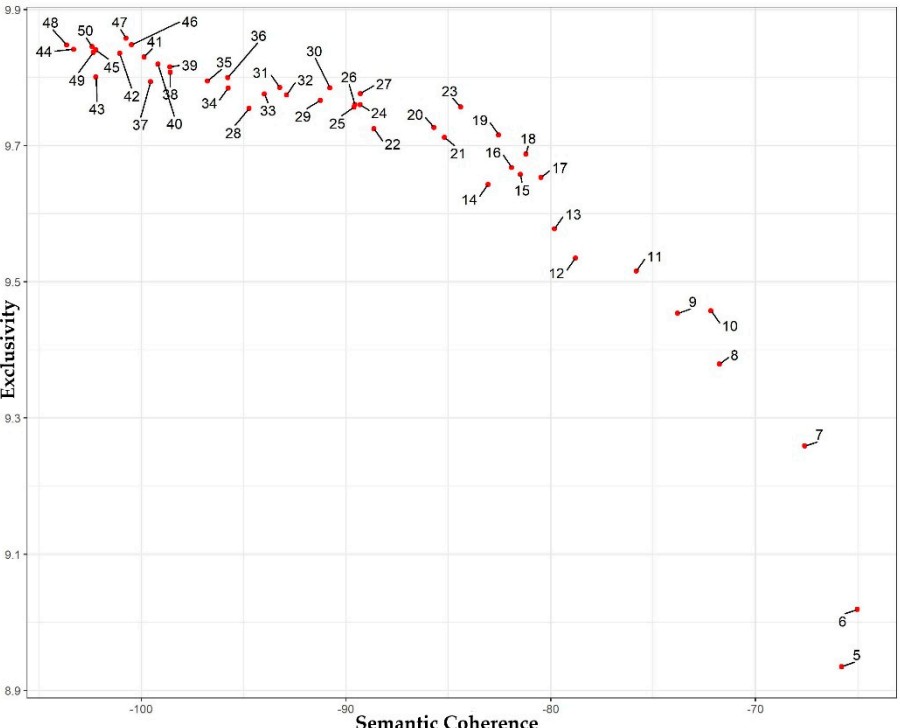

**Figure 2.** The plot of Semantic Coherence versus Exclusivity.

### 4.3. Identified Topics

Different terms are associated with specific topics following the application of STM. The exact label for the topic was based on the intuitive interpretations of topics conducted manually by the authors. The authors deduced the meaning of the themes from the frequency with which particular words appear within them. This was performed by meticulously evaluating each topic and its related collection of words. For example, the phrases develop, review, theory, and literature refer to the same topic. Therefore, we concluded that the topic is about literature research as per our topic labels.

The previous paragraph purposefully maintains that the possibility of labeling phrases associated with a topic is not a hard rule. The easiest method would be to utilize an STM

model to determine which words have the greatest probability (Highest Prob metric) of occurrence depending on the subject. The difficulty is that some phrases will emerge as high-probability terms for a broad array of subjects that are widespread yet have little meaning when used out of context. To overcome the difficulty of widespread subjects that have a high probability of occurrence, the STM package includes additional measures such as LIFT and FLEX. The LIFT metric is a commonly used metric that quantifies the probability of a word arising depending on the topic divided by the probability of a word happening throughout the corpus. This statistic identifies phrases that are much more abundant inside a subject than they are across the corpus. The difficulty with this metric is that unusual words are more likely to be highly ranked. Thus, it would be very difficult, if not impossible, to attribute an intuitive meaning to a broad topic purely based on the term's outsized importance within that subject. Bischof and Airoldi [65] propose the adoption of the FREX statistic to overcome this shortcoming. FREX measure is defined as the ratio of word frequency, and subject to word-topic exclusivity. To best comprehend the subject matter of our corpus, which is almost certainly diverse because it encompasses all types of research conducted in the management and organization, we used a combination of the Highest Probability, LIFT, and FREX metrics to assign intuitive labels to the subject matter. All three of the metrics mentioned above are used because there is no one right statistic. The labels for the topics and the words for all the three metrics, "Highest Prob", "LIFT", and "FREX" are in Table 3.

To further assess the significance of topics, we describe the twelve most prevalent topics, as measured by the percentage of papers dedicated to each. Figure 3 organizes the themes according to their highest probability words. For example, the "Decision Making" topic was discovered to be the most frequent research theme in the corpus. The following two most frequently discussed topics are "Literature Research" and "Motivational states". Other themes in order of topic proportions are "Relationships", "Psychology", "Organization Management", "Behavioral Science", "Therapeutics", "Social environment", "Academics", "Employment", and "Parent and children".

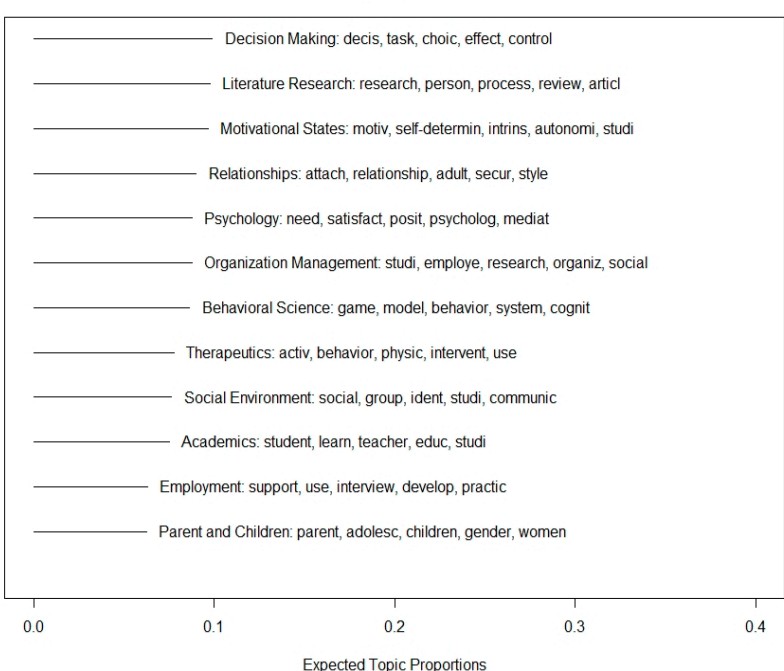

**Figure 3.** Topic Proportions of all the twelve themes.

**Table 3.** Topic Labels and associated words.

| Topic Label | Words (Showing Top Five Words) |
|---|---|
| Behavioral Science | Highest Prob: game, model, behavior, system, cognit<br>FREX: game, cooper, player, agent, evolutionari<br>Lift: con, game, metaphor, evolut, cooper |
| Academics | Highest Prob: student, learn, teacher, educ, studi<br>FREX: student, teacher, learn, academ, learner<br>Lift: entrepreneuri, profici, metacognit, classroom, learner |
| Psychology | Highest Prob: need, satisfact, posit, psycholog, mediat<br>FREX: satisfact, well-, mediat, job, need<br>Lift: harmoni, ill-, bootstrap, frustrat, work-rel |
| Relationships | Highest Prob: attach, relationship, adult, secur, style<br>FREX: attach, secur, infant, caregiv, bowlbi<br>Lift: ambival, hazan, aai, preoccupi, blehar |
| Social Environment | Highest Prob: social, group, ident, studi, communic<br>FREX: media, ident, self-control, comparison, communic<br>Lift: robot, outgroup, brand, offlin, intergroup |
| Organization Management | Highest Prob: studi, employe, research, organiz, social<br>FREX: organiz, exchang, leadership, organ, employe<br>Lift: top, citizenship, corpor, organiz, transact |
| Decision Making | Highest Prob: decis, task, choic, effect, control<br>FREX: decis, choic, task, prospect, probabl<br>Lift: busemey, gambl, tverski, monetari, Kahneman |
| Motivational States | Highest Prob: motiv, self-determin, intrins, autonomi, studi<br>FREX: intrins, motiv, autonom, sport, sdt<br>Lift: pelleti, amotiv, vallerand, sport, extrins |
| Parent and Children | Highest Prob: parent, adolesc, children, gender, women<br>FREX: gender, men, american, sexual, adolesc<br>Lift: bulli, delinqu, masculin, socioeconom, aggress |
| Therapeutics | Highest Prob: activ, behavior, physic, intervent, use<br>FREX: intervent, physic, activ, eat, food<br>Lift: condom, veget, diet, transtheoret, sedentary |
| Employment | Highest Prob: support, use, interview, develop, practic<br>FREX: career, interview, profession, qualit, care<br>Lift: mentor, themat, transcript, semi-structur, career |
| Literature Research | Highest Prob: research, person, process, review, articl<br>FREX: moral, patient, review, treatment, clinic<br>Lift: virtu, moral, ill, philosoph, psychodynam |

*4.4. Correlation between Identified Topics*

We examine the topics inside documents to determine the chance that a single document discusses a given group of topics. A lower distance between nodes and the presence of linkages in Figure 4 indicates a greater possibility that related topics are discussed in the same article. Correlation coefficients greater than 0.01 are used to create ties between topics, with correlation coefficients less than this value set to zero. Except for "Literature Research" and "Employment" other topics are associated with at least a topic even for this very low correlational level. This illustrates the fact that many research publications include connections between the study topics, implying that the research is multi-directional and multidimensional.

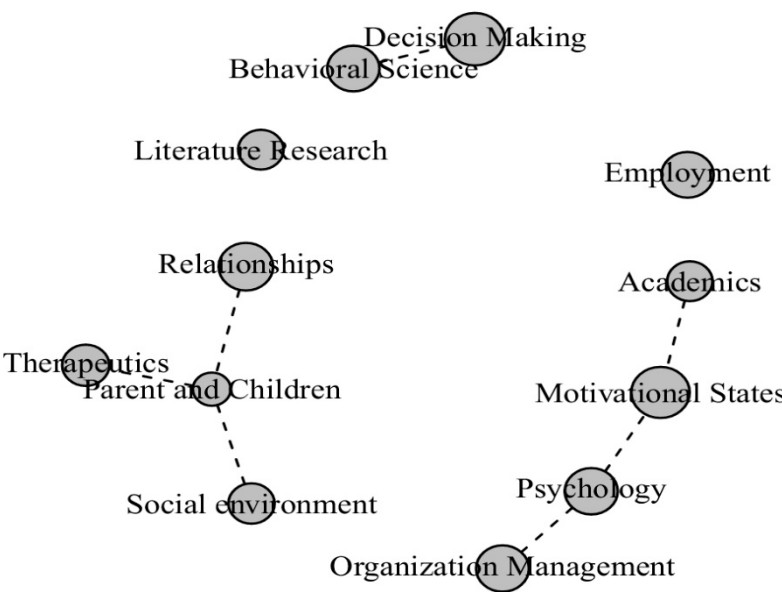

**Figure 4.** Correlation between topics.

Additionally, it is worth noting that "Literature Research", the second most-often occurring topic in the corpus, is separated from other topics. A topic such as "Literature Research" not having a correlation with other topics shows that the research on management literature or on the management and organization theories itself, rather than their application, is not multi-directional and multidimensional. In addition, topics such as "Employment" not having a correlation with other topics is quite surprising. It is noteworthy to see the correlation between "Academics", "Motivation", "Psychology", and "Organization Management". Similarly, topics "Parent and Child", "Therapeutics", "Social Environment", and "Relationships" are correlated. Furthermore, one more group of correlation between "Behavioral Science" and "Decision Making" is as expected due to the closeness of the topics in general.

Findings from the correlation of topics make intuitive sense. They justify the grouping of various themes of our STM model. They also help us to understand the gaps and opportunities available for research.

*4.5. Impact of Covariates on Topic Prominence*

The structural topic model investigated here utilizes document information to understand when occurrences happened to produce "document specific" estimations of topic prevalence. As a function of time, Figure 5 depicts the predicted subject proportions for all the topics. The prominence of the chart suggests a strong pattern for the topic over time. The proportion of topics "Behavioral Science", "Relationships", "Social environment", "Decision Making", "Parent and Children", and "Literature Research" has decreased with time in our dataset, while the proportion for topics "Academics", "Psychology", "Organization Management", "Motivational States", "Therapeutics", and "Employment" has increased. This is an important finding which provides excellent insights and opens various research opportunities. It shows the research on fundamental topics such as "Behavioral Science", "Decision Making", and "Literature Research" is decreasing at alarming rates. It is high time that researchers understand this and focus on the core research in addition to the application-based research covered in other topics. The exception to the above is regarding "Organization Management" and "Psychology" which are the core topics, and which still show an increase in the proportion of topics with time. A reason for this may be that the corpus was based on psychology papers. However, there is no obvious, intuitive explanation for this outcome.

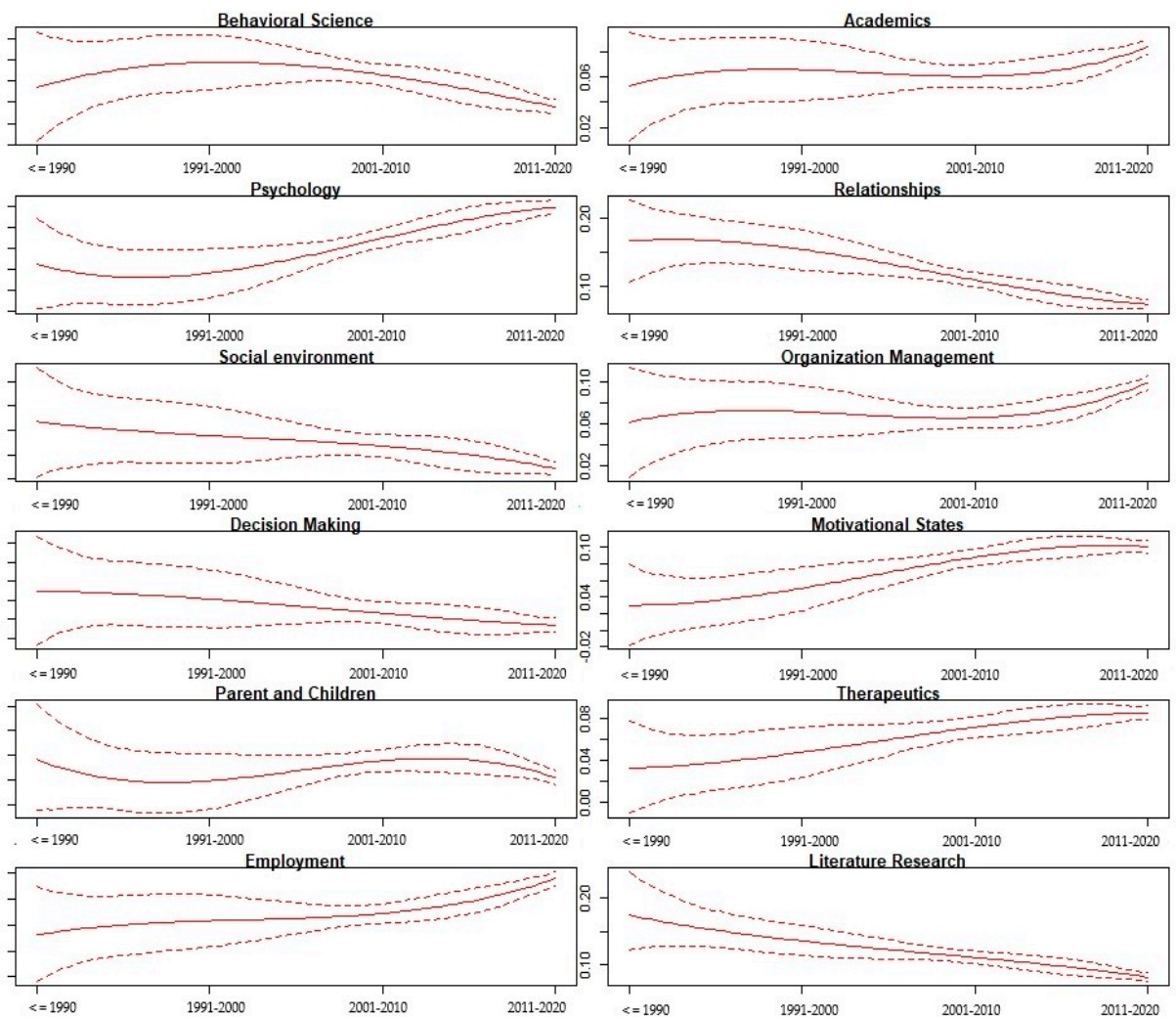

**Figure 5.** The prevalence of topics as a function of time is plotted with a 95% confidence interval.

## 5. Discussion

### 5.1. Principal Findings

According to the STM study, the theoretical application is not consistent across research topics. For instance, the topics indicate that the majority of theoretical applications are related to the top three study themes of "Decision Making", "Literature Research", and "Motivational States". Except for "Literature Research" and "Employment" all other topics are connected with at least one other, despite the correlation's low level. This demonstrates how often research publications incorporate linkages between study topics, demonstrating that the research is multi-directional and multidimensional.

The findings of this study reveal that the majority of research themes in management and organization are more concerned with discovering phenomena than with understanding and forecasting them. It looks as though the theory is employed only for its descriptive and communicative properties.

Examination of the evolution of the research areas over the years (see Figure 4) suggests that although many aspects of the management and organization research core remained stable over time, there were various notable changes. For example, topics such as "Academics", "Psychology", "Organization Management", "Motivational States", "Therapeutics", and "Employment" have become more researched with time. This reflects that the focus on each of these research areas is becoming prominent. At the same time, some research themes such as "Behavioral Science", "Relationships", "Social environment", "Decision Making", "Parent and Children", and "Literature Research" have become less

prominent. Such a decline may reflect the decreasing relevance of such topics, the fact that these topics have already been extensively researched, or that more finely grained investigations (using more precise terminology) are occurring later in a topic's life cycle.

The strength of the presented work is the use of unique analytical techniques to investigate the content of management and organization research literature, how topics are grouped, and how topics evolve over time. This outcome assessment is critical for evaluating how the study is progressing, the areas that need further attention, and some implicit discoveries. An additional study, however, is necessary to understand and build on the results given here.

### 5.2. Limitations and Future Scope

Our findings were based on the validity of the data we examined. We used a dataset that had been annotated and reviewed by multiple individuals to ensure that the annotation was of a high quality and that there were no errors in the annotations or calculations. Additionally, we repeated the study trials to confirm that no mistakes have been made and that our findings were reproducible. The forty management and organizational theories explored in this study are extremely practical for resolving management and organizational issues. They are based on extensive research and have proven to be practical and successful for organizations. As a result, our theory selection process ensures that the study is both relevant and practical.

We have only considered articles with at least ten citations. This removes unwanted noise from the data used for STM model. Correlation coefficients greater than 0.01 are used to create ties between topics, with correlation coefficients less than this value set to zero. The correlation results are logical, and correlation of most topics to other topics even for this very low correlational level validate our findings. We also depicted the subject proportions for all the topics with time. This helped to understand how the aspects of the management and organization research changed with time.

All these measures reduce the risk of unwanted distance between the results presented in the research article and the practical relevance. Thus, this study has relevant implications for future academic research as well as managerial practice. However, in future research, we would want to create a larger dataset to include research articles from multiple domains other than psychology. This would provide us with more flexibility in approaching our research question. We want to create a corpus of papers from various domains. It would aid us in determining the applicability of ideas in a larger context.

### 6. Conclusions

In this paper, we attempted to facilitate the identification of the application of theory in management and organization research by examining the vast body of academic research. We empirically identified key research areas and themes. This structural topic modeling study revealed twelve themes of research using theories for academic research on management and organization research literature in the psychology domain. The findings identified the research themes related to each other in terms of theory usage and revealed the change of themes with time. The analysis provided will serve as a guide to future study options. This study is a pioneering effort to gain insights into the intellectual core of management and organization research through the perspective of theory application. The findings of this research will contribute to the existing body of qualitative and quantitative studies on management and organization research.

**Author Contributions:** Conceptualization, R.B.M. and H.J.; formal analysis and writing—review and editing, R.B.M.; data curation and writing—original draft preparation, R.B.M.; supervision, H.J. All authors have read and agreed to the published version of the manuscript.

**Funding:** This research was funded by the National Natural Science Foundation of China, Project no. 71801195.

**Institutional Review Board Statement:** Not applicable.

**Informed Consent Statement:** Not applicable.

**Data Availability Statement:** Not applicable.

**Acknowledgments:** The authors would like to thank the anonymous reviewers for providing their constructive comments.

**Conflicts of Interest:** The authors declare no conflict of interest.

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
