# Peer review of "Management and Organizational Research: Structural Topic Modeling for a Better Understanding of Theory Application"

_sustainability, doi:10.3390/su14010159_

Round 1
Reviewer 1 Report
The subject of article is relevant for management science under the pressure of modelling and simulations, applied extensively nowadays.
My observations are related to the passing from reality to representations (as base of models) . Are reflected management theories reality by your mathematical models and evidences? Are corelated your results, in terms of robustness, with evolution of current and past management science theories ? The risk is to obtain a unwanted distance between theory and results, with relevant implications for practice.
Please develop extensively this idea at the section 5.2.
I would like also that figure 3 to be redesign in order to express at a high level of accuracy your intention.
Reviewer 2 Report
I reviewed the Manuscript ID (sustainability-1500432), entitled " Management and Organizational Research: Structural Topic Modeling for a better understanding of theory application" for Sustainability. The idea is interesting, and it has significant implications for management and organizational research. The paper's argument is built on an appropriate base of theory, concepts, and ideas. The results were presented clearly and analyzed appropriately. Although there are a few issues that must be handled in order for the manuscript to improve.
- The authors should try to explain the abstract more accurately. For example, in order for the abstract to be understood by the reader, it should clearly present the purpose, methodology, and results.
- If possible, it is preferable to provide a detailed experimental flowchart so that the proposed method can be easily referenced by other researchers.
- Figure 1 should be displayed in a more appealing manner.
